# Ten Key Insights into the Use of Spinal Cord fMRI

**DOI:** 10.3390/brainsci8090173

**Published:** 2018-09-10

**Authors:** Jocelyn M. Powers, Gabriela Ioachim, Patrick W. Stroman

**Affiliations:** 1Centre for Neuroscience Studies, Queen’s University, Kingston, ON K7L 3N6, Canada; jocelyn.powers@queensu.ca (J.M.P.); ioachim.gabriela@queensu.ca (G.I.); 2Department of Biomedical Sciences, Queen’s University, Kingston, ON K7L 3N6, Canada; 3Department of Physics, Queen’s University, Kingston, ON K7L 3N6, Canada

**Keywords:** spinal cord, fMRI, human, animal, methods, analysis, review

## Abstract

A comprehensive review of the literature-to-date on functional magnetic resonance imaging (fMRI) of the spinal cord is presented. Spinal fMRI has been shown, over more than two decades of work, to be a reliable tool for detecting neural activity. We discuss 10 key points regarding the history, development, methods, and applications of spinal fMRI. Animal models have served a key purpose for the development of spinal fMRI protocols and for experimental spinal cord injury studies. Applications of spinal fMRI span from animal models across healthy and patient populations in humans using both task-based and resting-state paradigms. The literature also demonstrates clear trends in study design and acquisition methods, as the majority of studies follow a task-based, block design paradigm, and utilize variations of single-shot fast spin-echo imaging methods. We, therefore, discuss the similarities and differences of these to resting-state fMRI and gradient-echo EPI protocols. Although it is newly emerging, complex connectivity and network analysis is not only possible, but has also been shown to be reliable and reproducible in the spinal cord for both task-based and resting-state studies. Despite the technical challenges associated with spinal fMRI, this review identifies reliable solutions that have been developed to overcome these challenges.

## 1. Introduction

The spinal cord is an important signal-processing center within the central nervous system (CNS), and is critical to many functions. However, research into spinal cord functions, and clinical assessment of its function, presents many challenges, and has been somewhat limited in humans due to its inaccessibility. Functional magnetic resonance imaging (fMRI) is an invaluable, non-invasive neuroimaging tool that enables the study of neural activity in the CNS, including the spinal cord. While fMRI is common practice for studies of brain function, spinal cord fMRI methods have been slower to develop.

The body of literature dedicated to spinal fMRI has been steadily expanding over the last 22 years, since its inception in 1996, with the bulk having been produced within the last decade (Figure 1). For the purposes of this review, a general search was conducted in PubMed (https://www.ncbi.nlm.nih.gov/pubmed/) with the terms ‘“spinal”, “fMRI”, “functional MRI”’, which produced 108 results (Figure 2). In addition to these, we supplemented 24 papers that were identified through external literature reviews. Of the 132 total records, 18 were excluded as they did not fit within the scope of this review.

Although the body of literature on spinal fMRI is steadily growing, there are common misconceptions regarding the use, reliability, and replicability of spinal fMRI as a research tool. The objective of this paper is to not only serve as a comprehensive review of the literature to date on spinal cord functional MRI, but also to put this body of work into perspective and suggest future applications. We have compiled 115 papers that include methods development, task-based and resting-state spinal fMRI, functional imaging in animals, and previous reviews of the literature. From this, we have broken them down into categories based on two main acquisition methods, field strength, and study condition (Figure 3).

In this review, we discuss 10 key topics that encompass and explain the current knowledge within the field. We will begin with a discussion of the applications of this technology, including animal models, and their place within the body of spinal fMRI research, and then a review of the studies that have been done in healthy participants and patient populations, both in the resting-state and with task-based paradigms. This is followed by an examination of the different spinal fMRI methods, including study design, data acquisition, the challenge of random and physiological noise, temporal and spatial resolution, and then spatial normalization methods and anatomical templates. This review will culminate in a summary of the progress that has been achieved, and a discussion of how spinal cord fMRI may advance in the future.

## 2. Animal Studies—Linking Neural Activity to BOLD Responses

Animal studies have proven to be essential to the development and validation of spinal fMRI. While much of the animal spinal fMRI research has been conducted with rats [1,2,3,4,5,6,7,8,9,10,11,12,13] there have also been cases of mouse [14], non-human primate [15], and feline models [16]. The smaller physical dimensions of the spinal cord present some challenges for fMRI methods, given the need for finer spatial resolution. However, the dimensions of the cord do not scale with body size, and the cross-sectional dimensions of the rat cervical spinal cord are roughly 3 mm × 5 mm, compared to roughly 10 mm × 16 mm in humans [6,9,17]. The need to anesthetize animals presents both challenges and advantages. While anesthetic agents may alter some functions and affect fMRI results, they also allow for longer time-series acquisitions and reduce bulk motion, thereby allowing for greater statistical power and less noise.

The necessity of animal studies is that they enable researchers to model disease or injury and obtain data that would not be possible from studying humans. These studies have involved either the study of normal neural processes [7] or the study of experimental peripheral nerve or spinal cord injury (SCI) [1,5,14], and have employed a range of stimulation methods including tactile stimulation [7], electrical stimulation [1,2,5,6,7,9,11,12,13,16], heat pain [4,15], or direct injection of capsaicin [8,9] or formalin [10]. Another advantage over human research is that longitudinal studies are much more feasible. For example, functional imaging sessions may be done multiple times over weeks or months, depending on the species, within a drug/treatment or disease model. In addition, the animals could then be used for histological studies at the conclusion of functional scanning, in order to further examine the underlying physiological processes.

Some of the key findings, to date, from animal studies, include a comparison of BOLD fMRI responses during electrical stimulation and histology with c-fos staining to identify regions responding to nociceptive input [2]. This study provides evidence of the link between BOLD responses and neural processes in the spinal cord. Other studies include investigations of the effects of diabetes and spinal cord injury in rats, and demonstrated differences in both responses with disease/injury conditions [1,7]. Functional MRI in the spinal cord has also been demonstrated in non-human primates, and showed differential BOLD responses to noxious heat and touch stimuli [15].

All of the animal studies, to date, have been important in establishing spinal cord fMRI methods and relating them back to the physiology of the tissues [2,10,12]. Much of the data obtained has allowed the field of spinal fMRI to progress in humans, as it has demonstrated that the spinal cord is a reliable source for BOLD contrast [2,3,4,5,6,7,8,9,10,15]. Despite the small physical dimensions of the spinal cord, magnetic field inhomogeneities caused by the surrounding vertebrae, and motion artefacts due to faster heart rates, the work in animal models has demonstrated that spinal cord fMRI is a viable method at a variety of magnetic field strengths: 1.5 T [14], 3 T [16], 4.7 T [1,10,11], 7 T [2,3,4,7,12,13], and 9.4 T [5,6,8,9,15]. The validation of the methods and confirmation of the link between BOLD responses in the cord and neural activity can be translated to human studies, although the latter are typically carried out at lower fields and with coarser resolution, as discussed in the sections below.

Although there are many strengths to spinal cord fMRI in animal models, and the methods have been replicated and validated, the insights gained about neural physiology may not be directly transferrable to humans in all cases. As Craig described in his 2003 paper, there is an evolutionary discrepancy between neural networks in lower animal species and those with an evolved neocortex, particularly between regions of interoception, homeostasis, and autonomic regulation [18]. It is clear that humans possess neural connectivity and capability that is far more complex than in some animal models (non-human primates being the exception), and therefore, there is a disconnect when attempting to translate results between species. While animal studies have been invaluable for establishing spinal fMRI methods and showing the reproducibility and reliability of detecting neural activity in the cord, we should be wary of generalizing behavioral or neural connectivity results to humans.

## 3. Applications in Healthy People—Demonstrating Validity and Versatility

Spinal fMRI has been used to demonstrate activity in the human lumbar, thoracic, and cervical cord in response to a variety of tasks and stimuli. Early spinal fMRI studies focused on basic sensory and motor tasks to explore whether activity that would be expected to occur in certain areas of the cord in response to these tasks (based on known physiology and neuroanatomy) could be reliably detected with spinal fMRI techniques. Functional changes in the cord in response to opening and closing of the hand were demonstrated as early as 1996 [19], and BOLD signal changes were demonstrated in the cord in response to hand closing and sensation of a cold stimulus in 2001 [20]. Many studies followed these first steps and successfully showed differences in BOLD signals in the spinal cord in relation to different intensities of touch or pressure stimuli [21,22], different types of sensory stimuli on the body [23,24], and various types of thermal stimuli [25,26,27,28,29,30]. Importantly, the results of these studies were very consistent over the years, and serve to demonstrate the reliability of spinal fMRI as a tool for investigating spinal cord activity in humans in a variety of scenarios.

The topic that has so far been given the most attention with spinal fMRI studies is that of pain processing and descending modulation of pain. This topic has broad applications, considering chronic pain conditions and strategies for pain management, and is also an area of research that benefits greatly from improved technologies and methods for observing spinal cord activity in humans in a non-invasive manner. Studies focused on pain processing reliably showed spinal cord activity in response to painful stimuli on various areas of the hands, arms, and feet [24,31,32,33,34,35,36,37,38,39]. These results confirm that spinal fMRI can detect spinal cord activity in response to pain, as well as localize this activity to specific spinal cord segments corresponding to the dermatomes being stimulated.

Several studies have also shown that BOLD signal activity measured with spinal fMRI can distinguish between different intensities of a painful stimulus (such as varying temperatures of a thermode applied to the skin) [31,33,34,40], as well as distinguish between painful and non-painful stimuli to the same area [24,32,34]. This body of evidence not only shows that different intensities of painful stimuli can be distinguished with spinal fMRI, but that individual differences in the perception of pain are also linked to BOLD signal changes in the spinal cord. Expanding on this knowledge base, Rempe et al. [41] showed that the response in the cervical cord and brainstem changed with sensitization of the skin (evidence for secondary mechanical hyperalgesia), while Bosma et al. [42] showed greater BOLD responses in the brainstem and cervical cord with temporal summation of second pain (corresponding with electrophysiological evidence derived from cat studies) [43]. This shows that spinal fMRI has advanced far beyond simply showing activity in the spinal cord in response to a stimulus. In addition, Khan et al. [44] found that individual differences in pain ratings of a stimulus corresponded to differences in BOLD signal changes in the brainstem and cervical cord. Studies have also shown that attention focused on or away from a stimulus can alter both the perception of pain and the corresponding BOLD signal changes in the cord. For example, diverting attention away from a stimulus generally elicited lower pain ratings and decreased input to the cord [45,46,47].

These findings link to those of broader studies showing descending modulation of pain in the brainstem and spinal cord [31,45,48,49]. Evidence that a state of threat or safety can alter this descending modulation of pain [48] has been supported by research that has focused specifically on placebo or nocebo effects using spinal fMRI [50,51,52]. Recently, Stroman et al. were able to show evidence of a continuous component to pain regulation in the brainstem and spinal cord [48,49]. This effect varies with individual pain sensitivity, and was demonstrated to occur throughout the duration of a study, not simply as a reaction to a painful stimulus, and could be linked to the changes in descending modulation of pain that have been observed in attention or placebo/nocebo-focused studies. Spinal fMRI has provided valuable evidence to advance the study of pain processing in humans, and will likely continue to have wide applications in the future, as studies continue to investigate how this evidence can translate to chronic pain conditions [53].

There are a variety of other topics that have been investigated, and which provide an expanding knowledge base on the organization and function of the human spinal cord. Recent studies, for example, have described BOLD signal changes in the thoracic and lumbar cord related to female and male sexual arousal [54,55], providing the first evidence of mapping sexual and autonomic function in the human cord, as well as showing that these are integrated functions that cannot be separated [55]. Studies have also employed emotionally valenced stimuli with spinal fMRI, showing BOLD signal changes in the cervical cord in response to negative mood images [56], BOLD signal changes in the thoracic cord related to whether participants viewed neutrally or negatively valenced images [56], and responses in the thoracic cord related to people making facial expressions of disgust as opposed to non-emotional expressions [57]. Interestingly, Smith et al. [56] also showed differences in BOLD signal changes in the cervical cord related to whether or not participants were shown images depicting people in motion, which may be due to motor priming or a perception of threat in certain situations.

The studies, to date, in healthy human participants have thus demonstrated the reliability and effectiveness of spinal fMRI for studying spinal cord function. The results have already provided considerable new insight into how the spinal cord functions, and its role in a number of processes. The spinal cord is clearly much more than a relay point between the periphery or organs and the brain.

## 4. Applications in Patient Populations—Broadening Our Understanding

The spinal fMRI studies that have been carried out in healthy participants have provided a basis for studies of spinal cord injury and disease. Stroman et al. [58] were the first team to use spinal fMRI in a patient population, showing that people with complete SCI who reported no sensation when thermal stimuli were applied still showed BOLD signal changes in the cord in response to these stimuli. While this activity differed from healthy controls, participants with incomplete injuries had remarkably similar activity in the cord to healthy controls, even though they reported feeling little sensation or altered sensation in response to the stimulus. This study was an important developmental step for SCI research and for the use of spinal fMRI in patient populations in general. Spinal cord fMRI has also been used to demonstrate that significant activity in response to motor or sensory stimuli is retained below the level of injury [59,60,61,62,63]. More recently, activity in the cord associated with female sexual arousal, in both patients with complete and incomplete injuries, has been characterized [54].

While SCI is the most widely-studied patient population when using spinal fMRI, this technology has also been used to identify differences in spinal cord activity in response to different stimuli in multiple sclerosis [64], to characterize spinal cord responses to heat pain in fibromyalgia [53], and to show resting-state spinal cord activity differences in patients with cervical spondylotic myelopathy compared to healthy controls [65]. A recent case study was also reported in which the investigators examined activity in response to painful stimuli in a patient with referred pain [66]. This body of literature shows that spinal fMRI has been an invaluable tool for studying spinal cord activity in various patient populations.

## 5. Resting-State-Confirming Activity in the Absence of a Task

Resting-state fMRI studies have been conducted for a considerable time in the brain, but spinal cord fMRI studies have only branched into this field as recently as 2010 [67]. Although many spinal fMRI studies have employed study designs without a specific stimulus or task, these are not generally thought of as “resting-state” studies because their focus was often on the development of the methods rather than on investigating any coordinated activity in the spinal cord that occurs in the absence of any stimulus or task (i.e., when the participant is “at rest”). The first investigation into potential resting-state BOLD signal variations was conducted by Wei et al. [67] and showed evidence of coordinated networks in the spinal cord in the absence of a stimulus. However, these results also showed BOLD signal variations at the frequency of respiration of the participants, indicating that artefacts from physiological noise may have been a problem. Studies, to date, have continued to expand on this, showing reliable evidence that spontaneous BOLD signal fluctuations do occur in the spinal cord in the absence of a stimulus [67,68,69,70,71,72,73], and that these likely represent coordinated resting-state networks [67,68,69,74,75]. While these studies have demonstrated similar results over time, the acquisition of spinal fMRI data is affected by a number of technical challenges caused by the small cross-sectional dimensions of the cord and its movement within the spinal canal [76,77,78]. Some critics have cited this as a reason to question the validity of these results (more on this in the sections “Putting the Noise into Perspective” and “Spatial Resolution”). Recently, Harita et al. [79] examined the contributions of various sources of physiological noise in a resting-state spinal fMRI study. By comparing their noise removal methods with data obtained from cadavers, they were able to provide evidence that a large proportion of the physiological noise can be removed, leaving predominantly Gaussian noise. They were also able to confirm resting-state BOLD fluctuations in the spinal cord after noise removal, and therefore provided additional credence to past study results [79].

While the studies, to date, provide strong evidence for coordinated resting-state fluctuations in the spinal cord, and the methods appear to be effective, this area of study is quite new and further technical improvements are to be expected. Evidence has shown that acquisition methods based on fast spin-echo provide higher quality data and less distortion than gradient-echo EPI, which many of the resting-state studies have employed (more on this in the section “HASTE vs. EPI”). Recent studies [67,68,69,74,75] have investigated coordinated resting-state BOLD signals across larger sections of the spinal cord and have shown evidence that these signal fluctuations likely represent coordinated resting-state networks. Future studies should focus on exploring resting-state networks directly (rather than simply coordinated BOLD signal activity within a particular slice) and should consider employing acquisition methods based on fast spin-echo in order to obtain high-quality data.

## 6. Study Design—Adapting to the Environment

The design of a functional MRI study includes the type of stimulus or multiple stimuli, the conditions used for comparison (may be other stimuli or no stimulus), the timing of when different conditions are presented, and also the instructions/information presented to the participant and any responses recorded from the participant. This design influences the neural activity that can be detected and has a tremendous impact on the quality of the results that are obtained. An efficient study design allows researchers to achieve high statistical power, while also maximizing the BOLD contrast between the neural processes under study.

In order to obtain reliable results and minimize bulk motion artefacts, studies to date show that it is beneficial to train participants outside of the scanner so that they can become accustomed to the unfamiliar environment and any additional supports (i.e., respiration monitor). Part of this training can include a special focus on learning how to lay as still as possible while confined inside the bore of the magnet. Since complete relaxation of body muscles in a supine position is not natural, participants can be instructed on what to expect and how to keep still during the study. This can be aided with additional foam support in the scanner, as well as straps across the chest and abdomen, if necessary [19,38,72,80,81]. This is a critical consideration for spinal fMRI, as head motion is easier to prevent with restraints or padding than is bulk motion of the body. Visual displays during fMRI studies have also been shown to help with maintaining participants’ focus on the task or stimulus and to prevent fatigue or boredom [22,42,47,48,49,53,82]. Depending on the necessity of salience for a chosen task, researchers may also want to train participants to become familiar with the stimulus or task (i.e., motor task, thermal stimulation, etc.) [19,26,28,37,38,40,47,48,50,52,53,57,62,80,81,83]. For example, Dobek et al. trained their participants in two 1 h sessions on the expectation of a noxious thermal stimulus, with or without synchronous presentation of music, and rating this stimulus on a visual analog scale, to obtain reliable reactions to a predictable stimulus [47].

There are two main types of stimulus presentation paradigms for functional MRI studies: block (to detect sustained activity) and event-related (to detect transient activity), and fast event-related designs are a hybrid of these two. The choice of design depends on the neural function of interest; block design is the most commonly used in the papers reviewed, as it is the most efficient. Moreover, peak efficiency is achieved in block designs when equal amounts of time are spent in each condition [84]. With exception to the resting-state studies (as discussed in the “Resting-State” section) all but one of the reviewed studies used a block design. Figley and Stroman were the only group to use an event-related design, in order to characterize the hemodynamic response in the cord [85]. Each method has merits and pitfalls in relation to the neural activity that is studied, but they are both able to reliably show BOLD signal intensity changes during a task in relation to a rest period.

The statistical power obtained from functional MRI data increases with the number of volumes collected, but this number must be balanced with the sampling rate and duration of each fMRI run. A simple way to achieve high power is to collect data in long runs, with multiple sequential stimuli [22,24,37,39,41,52,83,86]. However, a potential issue with long fMRI runs is that human participants can lose attentional focus on the task at hand. Participating in a functional MRI study often requires laying inside the bore of the magnet for an hour or longer; during which time the task can lose salience as the participant grows more tired or bored. This problem is worsened when studies employ runs of 15 min or longer, with little or no interaction with the investigators in between. In a study by Kong et al., 18 participants each underwent a single 40 min run, during which 80 stimuli were presented [86]. Although the authors do not discuss this as a limitation to their study, it is possible that the stimuli would lose salience, and that participants may create more motion artefacts with no breaks in between scanning.

A solution to this is to collect data in multiple short runs, with a smaller number of stimuli presented in each one [23,28,33,34,36,38,40,42,44,47,48,53,56,58,59,62,63,82,85,87,88]. Murphy et al. showed that the minimum number of volumes that should be acquired is determined by the percent signal change that can be detected (effect size), the signal-to-noise ratio (*SNR*) of the fMRI data, and the desired significance threshold [84].
(1)$$N=\frac{2}{R\left(1-R\right)}{\left(\frac{erfc^{-1}\left(P\right)}{\left(TSNR\right)\left(eff\right)}\right)}^{2}$$

In this equation, *N* is the number of time points required, *R* is the ratio of time points in the task period, *erfc*^−1^ indicates the inverse complimentary error function, *P* is the statistical significance *p*-value, *TSNR* is the temporal signal-to-noise ratio, and *eff* is the effect size. Additionally, short functional runs within the same participant, of the same study condition (e.g., task or rest) can be concatenated or averaged to increase the number of total volumes and statistical power [24,37,44,47,48,49,53,61,80,81,82,87]. Combining multiple short runs also provides the advantage of allowing time between runs for participant responses, avoiding fatigue, etc.

The optimal functional MRI study design captures changes in BOLD contrast by collecting the greatest number of volumes per run while simultaneously balancing the chosen stimulus presentation method, length of acquisition periods and, importantly, the comfort and attention of the participant.

## 7. HASTE vs. EPI—Contrasting Quality and Quantity

The two most common data acquisition methods used in the papers included in this review are gradient-echo with echo-planar imaging spatial encoding scheme (GE-EPI), and variants of a partial-Fourier single-shot fast spin-echo (HASTE). Of the papers reviewed that report human studies, 20 used GE-EPI and 36 used HASTE. The most commonly stated reasons for using these methods are that GE-EPI provides the best temporal resolution, whereas HASTE provides the best image quality. As will be discussed in the section on temporal resolution, typical HASTE methods have a repetition time (TR) of 6–7 s, whereas typical GE-EPI methods have a TR of 2–3 s. This raises a key question—which of these methods provides the best sensitivity to neuronal activity in the spinal cord?

Both GE-EPI and HASTE methods can be indirectly sensitive to changes in neural activity as a result of BOLD changes in T_2_* and T_2_, respectively. Although T_2_-weighting is less commonly used than T_2_*-weighting for fMRI, it has been well-established since the early days of fMRI [89,90]. The echo time (TE) for optimal BOLD sensitivity is equal to the T_2_* value of the tissues for gradient-echo methods (roughly 25 ms in the spinal cord at 3 T), and is equal to the T_2_ values of the tissues for spin-echo methods (roughly 75 ms in the spinal cord at 3 T) [77]. The BOLD sensitivity of the two methods are expected to be approximately equal, as shown in Table 1.

The relative image quality between GE-EPI and HASTE methods depends on a number of factors, but the *SNR* can be estimated for typical image parameters as follows [92]:(2)$$SNR\propto \mathrm{voxel}\;\mathrm{volume}\;\sqrt{\frac{N_{X}N_{Y}}{\mathrm{BW}}}e^{-\mathrm{TE}/\left(T_{2}^{*}{\;\mathrm{or}\;T}_{2}\right)}\;\frac{1}{\sqrt{\mathrm{acceleration}\;\mathrm{factor}}}.$$

This equation applies if T_1_-weighting can be ignored, and a roughly 90° flip angle is used for both methods. The “voxel volume” is the three-dimensional volume of each voxel represented in the image, the values of N_x_ and N_y_ are the image acquisition matrix dimensions, and the “acceleration factor” refers to the parallel imaging acceleration factor. With the acquisition parameters used in the most recent methods, the spin-echo method is expected to have more than 2 times higher *SNR* than the gradient-echo method (Table 2). However, another important factor is the acquisition speed, because the number of volumes that are acquired to detect the BOLD responses influences the sensitivity. Rearranging the expression described by Murphy et al. [84] to estimate the number of volumes needed, we can estimate the effect size (i.e., the % BOLD change) that can be detected for a given number of volumes (N):(3)$$eff=\frac{erfc^{-1}\left(p\right)}{SNR}\sqrt{\frac{2}{N\;R\left(1-R\right)}},$$
where *eff* is the effect size, *p* is the statistical threshold used, the function *erfc*^−1^ is the inverse complementary error function, and *R* is the proportion of time spent in the stimulation condition, assuming a block design with only two conditions. For the purposes of this comparison, we can set *R* = 0.5, and *p* = 10^−6^, which corresponds to *erfc*^−1^(*p*) = 3.46. An estimate of the corresponding *t*-value is given by √2 *erfc*^−1^(*p*), which is equal to 5.0. Using these numbers, we can compare the relative sensitivities of the methods, in terms of the % BOLD signal change that can be detected with a fixed acquisition duration (Table 2).

These estimates show that the faster sampling of the gradient-echo EPI method offsets its lower *SNR* and improves its sensitivity, so that it approximately equals the sensitivity of the HASTE method. With either of these methods, the duration of the fMRI acquisitions can be increased to provide greater sensitivity, within practical limits. Ultimately, the number of volumes that is acquired influences the sensitivity for detecting BOLD responses, not the sampling rate [92].

The final factor to be considered when comparing the spin-echo and gradient-echo methods is the image quality that is provided, as shown in Figure 4. Examples of GE-EPI and HASTE spinal fMRI data are shown in published papers by Tinnermann et al. [52] (in Supplementary Material), and Dobek et al. [47], respectively. Spinal fMRI acquisitions with gradient-echo EPI methods most commonly employ axial slices, since they appear to provide better image quality. However, in sagittal views extracted from the volume spanned by axial slices it can be seen that the images are still severely spatially distorted, and the distortion depends on the rostral–caudal position. Slice-specific shimming has been shown to improve the image quality over that shown in Figure 4, but it does not eliminate the distortions [93]. Alternatively, parallel imaging methods have been shown to reduce distortions, but are at the expense of lower *SNR* [94]. In addition, it has been shown that magnetic field shifts caused by changes in lung volume produce an artifactual spatial shift of the spinal cord in images acquired with EPI methods. This shift varies over the respiratory cycle, and must be corrected before fMRI analyses can be applied [95]. The net trade-off of using EPI methods is that they provide higher temporal resolution at the expense of loss of spatial fidelity, less anatomical coverage, increased physiological noise, lower *SNR*, and lower BOLD sensitivity, compared to single-shot fast spin-echo methods, respectively.

## 8. Physiological Motion—Putting the Noise into Perspective

A key challenge for spinal fMRI is commonly cited as being the large amount of physiological noise that arises from the proximity to heart, lungs, and throat, the movement of cerebrospinal fluid (CSF) and blood, and the movement of the spinal cord itself within the spinal canal. The amount of physiological noise has been claimed as limiting the reliability of spinal fMRI [96]. All of these various sources are related to cardiac and respiratory cycles. In addition to physiological noise, there is random noise in the MRI signal as well. Optimal methods require a balance of spatial resolution, acquisition speed, and *SNR*. The relative amount of random noise can be reduced at the expense of lower spatial resolution or slower acquisition speed. However, the non-random physiological noise presents a greater challenge, and is typically reduced by applying data processing methods after the data are acquired.

Spinal cord motion has been described as having an amplitude of approximately 0.5 mm in the cervical regions, and a period of movement that matches the cardiac cycle [76,78,97]. However, the phase of the movement relative to the cardiac cycle (i.e., the time at which the displacement is at peak values) varies with position along the cord. In addition, the amplitude of the movement diminishes more caudally along the cord, and the lumbar cord is almost motionless (relative to the spinal canal). A method has been developed to retrospectively model the physiological noise, due to this motion based on the cardiac cycle, and was termed “RESPITE” [97]. The model noise terms are included in a general linear model (GLM) analysis to account for the physiological noise in the data. This approach is reported to increase the specificity by 5–6%, and the sensitivity by 15–20%, for detecting neuronal activity. Another approach was developed based on retrospective image correction (“RETROICOR”), and included terms to model signal variations at cardiac and respiratory frequencies, and mixed frequency terms [98]. The resulting optimal method was reported to include 37 terms in a GLM basis set. A similar, but data-driven approach was also described, with noise terms identified by means of independent components analysis, and was termed “CORSICA” [83]. Both RETROICOR and CORSICA were tested with GE-EPI data, and were reported to reduce the effects of noise, as evidenced by increasing the numbers of active voxels detected, but the effect was not quantified.

These approaches only provide marginal reduction of noise and improvements in the ability to detect neuronal activity. However, it raises the question that if physiological noise is as large as the data seem to suggest, then why do these methods not provide more of an improvement? Assessments of the contribution of noise are based on the assumption that all signal variations are noise if they do not correspond with predicted BOLD responses to a stimulus or task. However, recent resting-state studies have demonstrated that there are coordinated BOLD responses in the spinal cord in the absence of a task or stimulus [67,71,79,99]. It has also been demonstrated that there are systematic BOLD responses in the periods preceding, and following, a noxious stimulus when no stimulus/task is being applied [48]. These BOLD signal variations have been attributed to continuous top-down regulation of spinal cord neurons. As a result, signal variations that do not correspond with the timing of a task or stimulus cannot be assumed to be noise.

Physiological noise in fMRI data acquired with a HASTE sequence has been assessed in relation to respiratory motion, cardiac motion, variations in end-tidal CO_2_, and bulk motion [79]. The results showed that in the resting-state, 46% of the signal variance could be removed by means of physiological noise modeling. Bulk motion accounted for 19% of the total variance, cardiac-related motion accounted for 14%, nonspecific signal variations detected in white matter accounted for 10%, respiratory-related motion was 2.6%, and end-tidal CO_2_ variations were 0.7%. These findings correspond with the previous assessment that removing RESPITE terms from the data (based on cardiac motion) improved the sensitivity by 15–20%. In the same study, fMRI data were acquired in two cadavers, and the results showed that 68 to 43% of the signal variance in resting-state data from healthy participants can be attributed to physiological noise [79]. Combined, these results show that the bulk of the physiological noise in HASTE fMRI data can be removed by means of physiological noise modeling, and that the physiological noise accounts for roughly the same amount of signal variance as the random noise. This puts the noise problem into perspective.

However, the noise problem appears to be greater when GE-EPI methods are used, because of the greater contribution of respiratory-related noise. The assessments of noise in HASTE data showed little contribution from respiratory-related motion, whereas RETROICOR and CORSICA methods which were tested with GE-EPI data demonstrated the need to include respiratory-related terms [83,98]. Efforts to develop physiological noise removal methods for GE-EPI consistently demonstrate a contribution from signal variations at respiratory frequencies [83,94,98,100]. As a result, the challenge presented by physiological noise is greater for data acquired with GE-EPI methods than with HASTE. The temporal signal-to-noise ratio has been reported to be approximately 15 with GE-EPI data (after motion correction was applied), and approximately 50 for data acquired with HASTE [40,44,101], and simulations yield *SNR* estimates of 24 and 56, respectively (as discussed in the section “HASTE vs. EPI”).

## 9. Temporal Resolution—Balancing Sensitivity and Speed

The challenge of balancing image quality, *SNR*, spatial resolution, and temporal resolution is at the core of all fMRI acquisition methods. The two most commonly used methods, GE-EPI and HASTE, demonstrate two very different strategies to achieve this balance. Spinal fMRI with HASTE methods (or variations of single-shot fast spin-echo) have been reported with repetition times of between 4.8 s and 9 s, with values around 6–7 s being the most common [22,23,24,25,28,31,33,34,36,37,41,42,44,45,47,48,49,53,54,55,56,57,60,61,62,63,64,66,76,77,79,80,81,82,85,87,88,97,102,103]. In most cases, data are acquired in sagittal slices and cover a large volume of the spinal cord. In contrast, methods based on GE-EPI are reported with repetition times typically around 3 s or as low as 1 s, but a few examples have TR values as short as 250 ms [26,27,29,30,32,35,39,40,46,50,51,52,65,67,75,83,100,104]. The increased imaging speed comes at the cost of less volume coverage of the spinal cord, lower spatial resolution, lower *SNR*, and poorer image quality than has been demonstrated with HASTE methods [101].

These two approaches raise the question of what temporal resolution is needed to accurately and sensitively detect BOLD signal variations in the spinal cord? The canonical BOLD hemodynamic response function takes approximately 5–6 s to reach the peak signal change after an increase in metabolic activity, and roughly 7 s to return to baseline levels upon cessation of a task/stimulus [105]. As a result, for block-design studies of responses to tasks or stimuli, a repetition time of 6–7 s may not show the transition in BOLD signal between states, but will demonstrate the BOLD change. For resting-state studies, it is expected that BOLD signal variations in the spinal cord occur at frequencies of less than 0.1 Hz, as has been shown in cortical regions [106]. Again, a TR of 6–7 s would be expected to show the variation in BOLD signal but may not capture the timing of the transition.

Analyses based on temporal correlation or GLM provide comparisons of BOLD signal time-series responses, or comparisons of BOLD responses with predicted responses. These methods are based on point-by-point comparisons, and do not depend on the signal intensity changes between successive points. As a result, it has been demonstrated that the sensitivity of GLM analyses for block-design fMRI paradigms increases with greater numbers of data points, and is maximal with approximately equal numbers of data points sampled during each study condition, as discussed in the section on “Study Design” [84]. The simulations shown in Figure 5 illustrate the insensitivity of linear fitting to the order of the data, or the time between successive data points. This simulation shows that fitting a dataset to a model paradigm, using a GLM, gives identical results if the data point pairs are sorted based on their values. Moreover, a linear fit of a plot of measured values vs. predicted values gives the exact same result. The sensitivity does not depend on sampling data points with sufficient temporal resolution to trace the rising and falling signal during changes in study conditions, or on the numbers of task/stimulus and baseline periods.

The properties of linear fits and correlation calculations show why it is important that a sufficient number of data values (i.e., volumes) are acquired during each condition in the stimulation/task paradigm. They also demonstrate that it is not necessary to sample points during the transitions between conditions. The primary benefit of faster sampling is that it provides more data during a given study paradigm. More data provides greater statistical power for detecting BOLD responses. Thus, it is clear that a repetition time of 6–7 s is inadequate for event-related studies, but it is adequate for block-design paradigms, for resting-state studies, and analyses by means of GLM and connectivity analysis. A large number of studies, to date, demonstrate this fact.

## 10. Spatial Resolution—Establishing the Precision

The small physical dimensions of the spinal cord are often cited as one of the key challenges for spinal fMRI. The optimal balance of spatial resolution, time to acquire images, and *SNR*, with the need to identify small anatomical regions, is not the same in the spinal cord as in the brain. The most commonly used method for task-based human studies (40 of the papers reviewed) is a single-shot fast spin-echo (HASTE, SSFSE), and most of these studies acquired data in sagittal slices [17,22,23,24,25,28,31,33,34,37,41,42,44,47,48,49,53,54,55,56,57,59,60,61,62,64,66,70,77,79,80,81,82,85,87,101,102,107]. The spatial resolution was typically 1 × 1 mm^2^ to 1.5 × 1.5 mm^2^ in the sagittal plane, with 2 mm-thick slices. Two exceptions acquired data in axial slices with finer in-plane spatial resolution (0.39 × 0.39 mm^2^ or 0.9 × 0.9 mm^2^) with 7 mm-thick slices [21,80]. Of 20 papers reporting human studies using GE-EPI, 19 used axial slices and 1 used sagittal slices [26,27,29,30,32,35,39,40,46,50,51,52,65,67,75,83,100,104,108,109]. The resolution in axial planes was typically 1 × 1 mm^2^ to 1.5 × 1.5 mm^2^, with 4 to 5 mm-thick slices. A few exceptions had thicker slices as large as 10 mm, and a few had larger in-plane resolution up to 2 × 2 mm^2^.

Most of the spinal fMRI studies reported, to date, therefore acquired data with much finer resolution than is typically used for brain fMRI. However, the actual spatial resolution that is achieved with EPI methods depends on the amount of spatial distortion and the effects of methods used to correct the distortions. The methods based on fast spin-echo acquisitions are free of such distortions (as described in the section on HASTE vs. EPI). Data acquired with axial slices typically provides finer in-plane resolution and lower rostral-caudal resolution than is obtained with sagittal slice methods.

The results of studies of specific functions provide evidence of whether or not the spatial resolution currently achieved is sufficient to detect neural activity that is localized to specific anatomical regions in the spinal cord. Examples of studies employing fast spin-echo methods include a study in which axial slices were acquired with 0.9 mm in-plane resolution. The results demonstrated fine details of dorsal and ventral horn responses with cold thermal stimulation on the leg [58]. A similar method was used to show fine details of the cross-sectional distribution of function in people with spinal cord injury [59]. Anatomical details of both cervical and lumbar responses to vibration stimulation of a number of different locations overlying tendons (palm, wrist, biceps, knee, Achilles tendon) have also been demonstrated. The responding regions were shown to vary in the rostral–caudal position in the cord with the location stimulated, and to occur in specific locations within the cord cross-section [23]. Similar methods were used to show details of responses to finger-tapping, and showed predominantly ventral region activity [28]. Fine details of anatomical regions responding to repeated touch on the right hand with 15 g von Frey filaments, as compared to 2 g von Frey filaments, and also two brushes of different stiffness, were shown using data acquired with HASTE at a resolution of 1 × 1 × 2 mm^3^ in sagittal slices [24]. The results showed different magnitudes of BOLD responses that corresponded with ratings of sensation/pain, and specific right dorsal and left ventral regions responding within specific spinal cord segments (C6 to C8). More recent studies using HASTE methods with sagittal slices, 1.5 × 1.5 × 2 mm^3^ resolution, have shown anatomical details of locations responding to sexual stimulation in the lumbar spinal cord in both healthy participants and three cases of women with spinal cord injury [54,55,64]. In addition, a number of studies of pain processing with similar methods have shown detailed responses in the cervical spinal cord to noxious heat stimulation and pain modulation by individual differences [44], spinal cord injury [62], temporal summation [42], and music [47]. Each of these studies demonstrates the ability to localize specific spinal cord segments and cross-sectional locations within the butterfly-shaped gray matter regions. These are only a few selected examples to identify a range of functions.

Examples of specific anatomical regions identified with GE-EPI methods include a study of placebo effects with 1 × 1 mm^2^ in-plane resolution in 5 mm-thick axial slices that identified BOLD responses localized to the left dorsal quadrant at the level of the C5 vertebra, with thermal stimulation of the left volar forearm (C6 dermatome) [50]. This region had lower BOLD responses during induced placebo effect. Another study using similar acquisition and thermal stimulation methods appeared to identify the exact same region of the spinal cord, and BOLD signal reductions during attention modulation with a working memory task [46]. Notably, the same localized region at the C5 vertebral level was identified in a study of responses to painful thermal stimulation [39], and a nearby adjacent region (more midline) was identified as having BOLD responses that were increased by a nocebo effect [52]. All of these examples employed thermal stimulation of the left volar forearm. In comparison, a study of simple versus complex motor tasks with the left hand identified BOLD responses approximately one and two vertebral levels more caudal than the previous thermal stimulation responses [29]. However, the activity was not well localized to specific regions within the cord cross-section, and this study employed axial slices with 2.5 × 2.5 mm in-plane resolution and 4 mm-thick slices.

In comparison, two related recent studies, employing GE-EPI with 1 × 1 mm^2^ resolution with 3 mm-thick axial slices, compared responses to warm and noxious hot thermal stimulation [40], and identified activity spreading across the C3 to C7 vertebral levels. This activity was primarily dorsal but was on both sides of the spinal cord, and was not well-localized. In a separate study with similar methods, responses to right and left wrist flexion tasks were compared and, again, showed activity spread across 4 or 5 vertebral levels [30]. The authors showed more localized activity with more stringent statistical thresholds. The findings of distributed rostral–caudal activity are in contrast to other studies using similar GE-EPI methods, and may be a consequence of lower signal-to-noise ratio resulting from smaller voxels with 3 mm-thick slices.

Overall, these examples demonstrate a high degree of consistency across studies, and sensitivity to specific functions and manipulations such as with attention, music, placebo, pain, finger-tapping, vibration of specific locations, etc., which correspond with the known neuroanatomy. These examples support the conclusion that spatial resolution of 2 mm or smaller is sufficient to localize BOLD responses to specific anatomical locations with both fast spin-echo methods and GE-EPI methods. The results also support the conclusion that lower resolution can be used in the rostral–caudal direction with axial slices. Most of the examples cited have voxel volumes of 4.5 to 5 mm^3^, but it appears that data acquired with finer resolution and voxel volumes of 3 mm^3^ may suffer from inadequate signal-to-noise ratio.

## 11. Spatial Normalization and Anatomical Templates—Automating the Analysis

The basic concepts underlying data analysis of spinal fMRI are identical to those of brain fMRI. The challenges that are specific to spinal fMRI analysis include the problems of physiological noise (discussed in “Putting the Noise into Perspective”), spatial normalization, and the closely-related task of co-registering the data.

The ability to identify specific anatomical regions with accuracy, and to compare results across individuals and across complete study groups, depends on the ability to normalize data to a common anatomical template. Early efforts to spatially normalize spinal fMRI data identified the challenges of focusing the normalization on the spinal cord, as opposed to the surrounding spine anatomy, muscle, and other tissues. Normalization methods that have been developed for the brain have the advantage that the skull/scalp can be identified and removed from the data, and/or fat suppression is applied, so that the entire imaging volume can contain only brain structures of interest and surrounding areas with little or no signal. Moreover, the brain shape is fairly consistent across participants. In contrast, the spinal cord and surrounding regions can vary in shape and position with neck flexion and thickness of surrounding tissues.

Early normalization methods described for the spinal cord were based on initial manual identification of reference points or the cord cross-section [17]. The first spinal cord template was described in 2008, and was based on *T2*-weighted images from 8 healthy participants and spanned the cervical cord and brainstem [110]. The method involved identifying reference points and mapping the cord/brainstem to a space with one axis parallel to the long axis of the cord, a second axis right/left, and a third anterior–posterior. The method focused on ensuring that the length of the spinal cord was not altered by the normalization procedure and did not require that the vertebrae were aligned across participants. This decision was based on prior studies of cadavers that found that the lengths of spinal cord segments were more consistent across individuals than were the lengths of vertebral bodies [111]. A normalized template and corresponding anatomical region map were also defined with 1 mm cubic voxels. This template was the basis for the development of improved normalization methods, and a fully automated method for normalizing the brainstem and cervical cord was described in 2015, with a template based on 356 healthy participants [34,42,77]. The accuracy was characterized as having 97% ± 3% of voxels matching the template within 1.5 mm.

Another template was developed based on anatomical image data with multiple contrasts in order to enable segmentation of gray matter, white matter, and cerebrospinal fluid. The development progressed in a series of iterative templates beginning with manual definition of the cord cross-section and increasing in precision with each iteration. The normalization method was, again, based on primary axes along the cord long axis, right–left and anterior–posterior, but data were aligned based on vertebral levels. The development began in 2014 with the MNI-Poly-AMU template, and probabilistic atlases based on 8 participants [112], followed by the AMU15 template (based on 15 participants) [113], the AMU40 template (based on 40 participants) [114] and, finally, the PAM50 template (50 participants) in 2018 [115]. The resulting templates span the entire length of the human spinal cord with 0.5 mm cubic voxels, and provide corresponding atlases of gray matter and white matter at all levels of the cord. These atlases demonstrate that 0.5 mm^2^ resolution is sufficient to identify variations in gray matter cross-sectional anatomy across participants and, thus, provides a high degree of spatial precision.

An important component of the spatial normalization is the ability to localize specific spinal cord segments which do not correspond with the vertebral levels. An anatomical study was carried out using high-resolution MRI in healthy human participants, and mapped the spinal cord segment positions based on the nerve roots [116]. The measured positions of the spinal cord segments are compared between this study and the previous cadaver work in Table 3. The reported lengths of the spinal cord segments are similar, but are generally larger in the study based on cadavers, and the distance between the C3 and C8 segments is similar between the two studies (relative to the length of a cord segment) at 64.6 mm and 71.3 mm. The difference between the measured segment positions and sizes could be the result of individual differences affecting the group averages in the two studies. However, the positions reported by the two methods have an average difference of 9.2 mm, and so they appear to be offset by one segment. The MRI-based measures begin at the 3rd cervical segment, and this difference raises the question as to whether the segments were misidentified by one position. The results of these analyses indicate that we are able to estimate the spinal cord segment position based on the distance along the cord/brainstem from the pontomedullary junction, with an apparent uncertainty of approximately 9 mm. Both of these studies of the segment positions and lengths conclude that the spinal cord segment positions in each individual cannot be identified based on the vertebral position, and that the position along the cord itself is a more accurate estimate [111,116].

The task of co-registering data to correct for small amounts of bulk body movement, changes in posture, etc., is essentially identical to the task of fine-tuning the spatial normalization to a template once the larger-scale coarse normalization has been applied. Thus, once an effective method has been developed for spatial normalization, the methods needed for co-registration are also in place.

## 12. Conclusions

The body of literature, to date, demonstrates that significant progress has been made toward establishing the validity of fMRI methods for studying neural processes in the spinal cord, in both humans and animal models. The challenges that are specific to spinal fMRI include the small physical dimensions of the spinal cord, the non-uniform magnetic field environment, and MRI signal variations caused by physiological motion. However, these problems have been largely overcome by adapting fMRI acquisition and analysis methods for data from this region. Animal studies have demonstrated the link between neural activity and BOLD responses in the spinal cord, and studies of healthy humans demonstrate the reliability and sensitivity of spinal fMRI. These studies span a wide range of functions, including motor, sensory, pain, and sexual function. Moreover, studies of patient populations have provided new insight into changes in spinal cord function in relation to traumatic injury, multiple sclerosis, and fibromyalgia syndrome. While most studies, to date, have employed block-design paradigms with tasks or stimuli, resting-state studies have also been carried out to show robust networks of coordinated regions in the spinal cord, even in the absence of any stimulus. These studies have relied on the development of effective methods for acquiring high-quality image data, removing physiological noise, and enabling analyses with automated identification of anatomical regions. After over 22 years of development, spinal fMRI methods have been well validated, and the more recent studies tend to be focused on applying fMRI to the study of spinal cord functions in injury and disease conditions, as well as to better understand healthy human physiology. Further improvements in the methods to produce sensitive and reliable results are still expected as our understanding of spinal cord physiology and anatomy improves, and, as a result, our understanding of what constitutes actual noise and artifacts improves. The results to date consistently demonstrate that the spinal cord is a dynamic system of ascending and descending signaling, to modulate and coordinate functions and maintain homeostasis, and is far more than a relay point for signaling between the brain and the periphery. While it is still yet to be determined whether or not spinal cord fMRI will ever be a practical clinical assessment method, there is ample evidence of its value as a research tool, and it may also prove to be useful for validating other clinical assessments.

## Figures and Tables

**Figure 1 brainsci-08-00173-f001:**
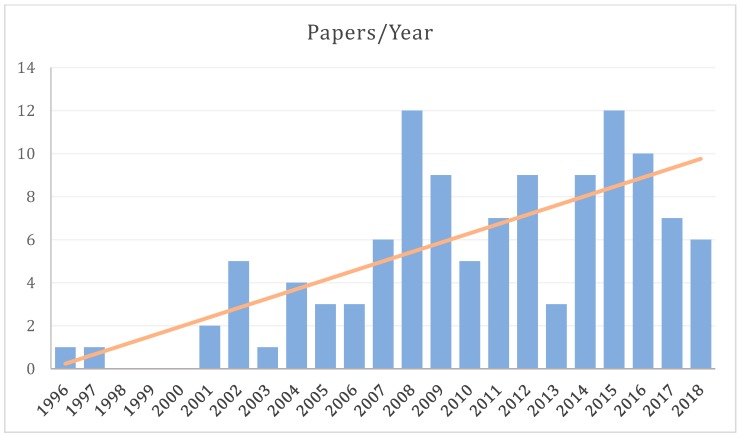
Histogram of the number of papers published per year on spinal fMRI, since its inception in 1996.

**Figure 2 brainsci-08-00173-f002:**
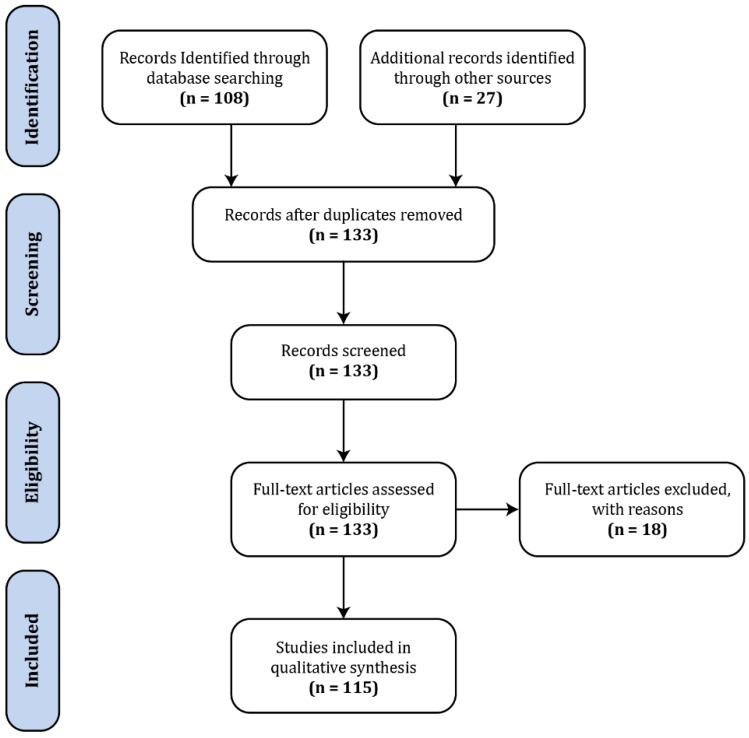
Flow chart of the literature review process.

**Figure 3 brainsci-08-00173-f003:**
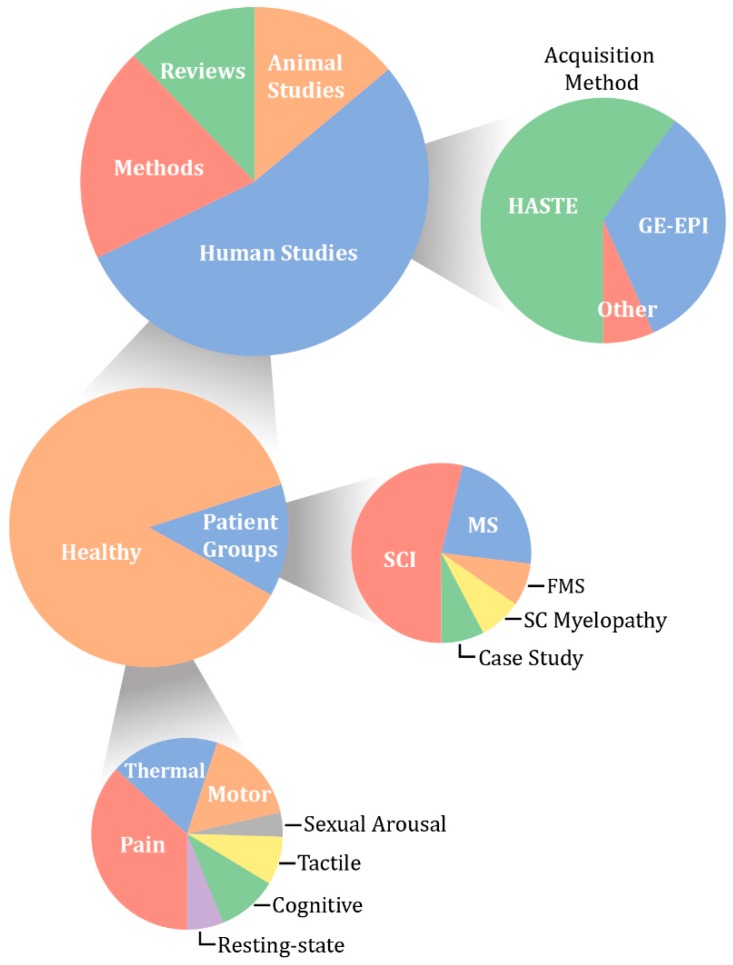
Chart of the distribution of spinal fMRI papers published to date, broken down by type, methods, study groups, and primary topic. HASTE—half-Fourier single-shot turbo spin echo; GE-EPI—gradient-echo with echo-planar spatial encoding; MS—multiple sclerosis; SCI—spinal cord injury; FMS—fibromyalgia syndrome.

**Figure 4 brainsci-08-00173-f004:**
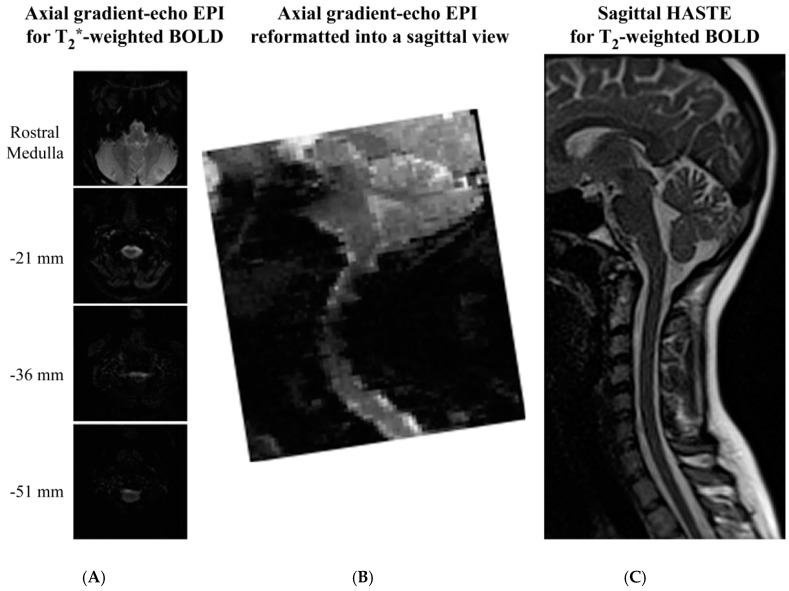
Comparison of image quality obtained with gradient-echo EPI and spin-echo HASTE sequences for spinal fMRI. Gradient-echo EPI images were acquired in contiguous axial slices (**A**) and were reformatted into sagittal views (**B**) for comparison with spin-echo (HASTE) images acquired in sagittal planes (**C**). Selected axial slices are shown for comparison, and the slice positions are indicated relative to the caudal medulla (top slice). Images were acquired at 3 tesla with a Siemens MAGNETOM Trio at Queen’s University, and used as examples for this review.

**Figure 5 brainsci-08-00173-f005:**
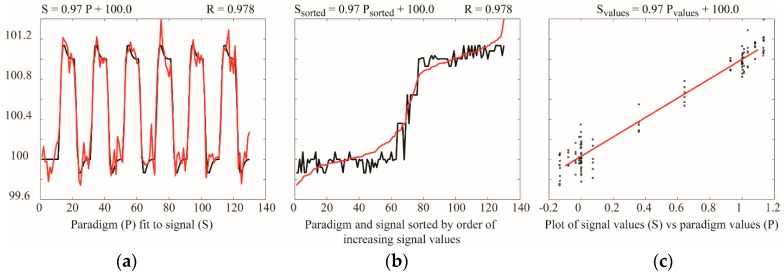
Simulations of data analysis concepts for the purpose of this review article. The left panel (**a**) shows a model paradigm, P (black), fit to a simulated signal, S (red), consisting of a 1% BOLD signal change from a baseline value of 100, plus random noise. The results of a GLM fit are shown: S = 0.97 P + 100.0. The correlation between the paradigm and signal is shown as *R* = 0.978. The middle panel (**b**) shows the effect of sorting both the paradigm values and the signal values to order the signal values in increasing order. The results of the GLM fit and correlation are unchanged. The right-most panel (**c**) shows a plot of signal values versus paradigm values, and a linear fit between them. The fit and correlation results are again the same.

**Table 1 brainsci-08-00173-t001:** Estimate of relative BOLD signal changes (percent change from baseline) with gradient-echo and spin-echo methods at optimal echo times.

Gradient-Echo	Spin-Echo
$\frac{\Delta S}{S}≅-\mathrm{TE}\;\Delta \left(\frac{1}{T_{2}^{*}}\right)$	$\frac{\Delta S}{S}≅-\mathrm{TE}\;\Delta \left(\frac{1}{T_{2}}\right)$
$\frac{\Delta S}{S}≅-0.025\;s\times -1.22\;s^{-1}$	$\frac{\Delta S}{S}≅-0.075\;s\times -0.37\;s^{-1}$
$\frac{\Delta S}{S}≅3.1\%$	$\frac{\Delta S}{S}≅2.8\%$

Values are shown for data collected at 3 T, in the spinal cord, with TE (echo time) values selected for optimal BOLD contrast [89,91]. T_2_: transverse relaxation time, T_2_*: apparent transverse relaxation time including effects of magnetic field inhomogeneity, ΔS: signal change between conditions, S: MRI signal intensity.

**Table 2 brainsci-08-00173-t002:** Estimates of *SNR* for commonly used spin-echo and gradient-echo spinal fMRI acquisitions. *SNR* values are estimated compared to a typical brain fMRI method, which is assumed to have an *SNR* of approximately 150 at 3 T.

	Typical Brain fMRI	Gradient-Echo Spinal fMRI	Spin-Echo Spinal fMRI
Imaging Parameters	3.3 mm × 3.3 mm	1 mm × 1 mm	1.5 mm × 1.5 mm
	3.3 mm-thick slice	5 mm-thick slice	2 mm-thick slice
	200 kHz bandwidth	200 kHz bandwidth	151 kHz bandwidth
	64 × 64 matrix	128 × 128 matrix	192 × 144 matrix
	TE = T_2_*	TE: 30 ms (~1.2 T_2_*)	TE: 75 ms (T_2_)
	acceleration factor = 1 (no parallel imaging assumed)	acceleration factor = 2	acceleration factor = 1
Estimated *SNR*	150	24	56
Acquisition time/volume	3 s	1.5 s	6.75 s
Detectable effect size, *p* = 10^−6^, 12 min acquisition	0.42%	1.9%	1.7%

**Table 3 brainsci-08-00173-t003:** Measured cervical spinal cord segment positions in millimeters from the pontomedullary junction (PMJ), and the lengths of each segment, measured based on anatomical MRI (Cadotte et al. 2015) [116] and from cadavers (Lang and Bartram 1982) [111].

	Cadotte et al. 2015 [116]		Lang and Bartram 1982 [111]	
	Middle of Segment Position	Length (mm)	Middle of Segment Position	Length (mm)
C1			25.0	8
C2			35.3	12.5
C3	51.5	10.5	46.7	10.4
C4	65.7	9.9	57.7	11.5
C5	81.1	10.5	70.9	15.0
C6	95.4	9.7	85.4	14.0
C7	109.3	9.4	98.6	12.4
C8	122.8	9.6	111.3	13.0
	Distance from C3 to C8	71.3 mm	Distance from C3 to C8	64.6 mm

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
