# Peer review of "Ten Key Insights into the Use of Spinal Cord fMRI"

_brainsci, 2018, doi:10.3390/brainsci8090173_

Round 1

Reviewer 1 Report

I ‘d like to congratulate the authors for this well written review on a very important and promising topic that hopefully will renovate interest and future applications.

This is a valuable tool for investigate many pathological traits of the spinal cord together with its physiology. This is a very well written and described paper bringing the reader to the knowledge of a very modern approach to investigate the mysteries of the spinal cord. I consider it the starting milestone to open a new avenue to investigate many spinal pathologies including spinal cord injuries, chronic neuropathic pain and stuff. Congratulations to the authors that were able to elicit a great interest in such a specialized branch of medicine that a few reader would reach the end of the paper without losing attention.

Author Response

We thank the reviewer for the time taken to carry out this review and their very positive comments.

Thanks!

Reviewer 2 Report

This review paper has been organized like a book chapter. Readers can gain general useful information from it, but it does not review previous literatures in detail. It has categorized information related to the subject but does not provide a sense to readers about the state of the art. For example, in line 300, authors have mentioned “A solution to this is to collect data in multiple short runs, with a smaller number of stimuli presented in each one”, then they have referred their readers to 21 references. After that they immediately have spoken about Murphy et al work which is a different reference. If the reason to have 21 references there is to show that statement was correct, 2-3 references were enough. If they wanted to show other researchers have used smaller number of stimuli method, was not it useful to explain each researcher method and their findings? Or at least explain the most important ones? This is not a literature review if authors just refer their readers to some references and provide just a general statement.  

I see this problem in several parts of the paper like lines 272, 310, 462, 466, 518, 523 …

I rarely see authors compare results of two researchers or show the advantage of one researcher method over the others.

Figures 4 and 5 need citations in their captions.

Author Response

We appreciate the time taken by the reviewer and the helpful comments. We have considered each point carefully as to how we can make our manuscript more clear and informative, and also more useful to the reader.

We believe that an effective review of the literature provides much more than a synopsis of the work that has been published, and provides new information by synthesizing the published work into what consistent findings can be taken from it. We reviewed 115 papers and so it is not practical or even useful to review each of these papers individually. The alternative is to select specific papers to review in detail, but it would be difficult to choose which papers to highlight. This would give the impression of “cherry picking” the literature, possibly to make conclusions that are biased toward our own interests. With the approach we have taken we provide a complete list of the published work that supports the overall conclusions that we draw from the entire body of literature, and we show that our assertions are well-founded.  Since the manuscript is already almost 9500 words long, we believe that making it longer by adding details about the methods or findings in each paper that we review would make this manuscript less useful. The purpose of our review was not to summarize individual research findings, but to use the findings of the 115 papers in this field to create a resource from which a reader can glean information about the most viable methods, varied applications, and state of the art knowledge in the field, based on the evidence available. We believe we have achieved this purpose. 

With regard to the specific example provided, we chose to cite the 21 papers which all employed multiple short runs, because we are making the claim that this method is well demonstrated and supported by published work. If we make the point that N papers showed something, then we should cite N papers so that the reader can find this body of work. The reference to Murphy et al. is in regard to the equation that we provide, and this is a separate point and a separate source being cited. Similar to our comment above, how would we choose “the most important ones”? If we choose the work from our own group as being the most important we would give the impression of being biased. Instead, we chose to give a thorough and fair review of what has been done in the literature. That being said, we also included specific examples from the literature to support specific points throughout the manuscript (we counted over 25 instances).  We believe that we have achieved the exact purpose of a review article, which is to synthesize the current literature into the main findings.

The third comment by the reviewer (“I rarely see authors compare …”) is somewhat ambiguous.  We are not sure if this is raised as a strength of our manuscript because we compare the work that has been published, or a criticism. In a review of the literature, we believe that it is a critical part of the review to compare the work that has been published. This is part of obtaining a sense of what has been learned so far from the body of literature.

Figure 4 was created by the authors. There is no reference to cite for this figure.  We have revised the figure caption to indicate the source of the images.

Figure 5 shows simulated data and was also created by the authors, and there is no reference to cite for this figure. We have clarified this in the figure caption.

Round 2

Reviewer 2 Report

This paper provides very useful information about the subject. Authors have reviewed 115 papers which is great. I was hoping authors could select few of these as must-read papers in the field and explain them more in detail, however, I was convinced by authors that they prefer not to add more information to their manuscript because this paper is already long. I recommend this paper for publication. Thank you.